# Prominent and Regressive Brain Developmental Disorders Associated with Nance-Horan Syndrome

**DOI:** 10.3390/brainsci11091150

**Published:** 2021-08-29

**Authors:** Celeste Casto, Valeria Dipasquale, Ida Ceravolo, Antonella Gambadauro, Emanuela Aliberto, Karol Galletta, Francesca Granata, Giorgia Ceravolo, Emanuela Falzia, Antonella Riva, Gianluca Piccolo, Maria Concetta Cutrupi, Pasquale Striano, Andrea Accogli, Federico Zara, Gabriella Di Rosa, Eloisa Gitto, Elisa Calì, Stephanie Efthymiou, Vincenzo Salpietro, Henry Houlden, Roberto Chimenz

**Affiliations:** 1Department of Human Pathology in Adult and Developmental Age “Gaetano Barresi”, Unit of Emergency Pediatric, University of Messina, Via Consolare Valeria 1, 98125 Messina, Italy; celestecasto@libero.it (C.C.); dipasquale.va@gmail.com (V.D.); gambadauroa92@gmail.com (A.G.); giorgiaceravolo@gmail.com (G.C.); cutrupimaricia@gmail.com (M.C.C.); 2Unit of Ophthalmology, Department of Clinical and Experimental Medicine, University of Messina, Via Consolare Valeria 1, 98125 Messina, Italy; ceravoloida@gmail.com; 3Casa di Cura la Madonnina, Via Quadronno 29, 20122 Milano, Italy; emanuela.aliberto91@gmail.com; 4Department of Biomedical, Dental Science and Morphological and Functional Images, Neuroradiology Unit, University of Messina, Via Consolare Valeria 1, 98125 Messina, Italy; karolgall@yahoo.it (K.G.); francesca.granata@unime.it (F.G.); 5Azienza Ospedaliera di Cosenza, Via San Martino, 87100 Cosenza, Italy; emanuelafalzia@yahoo.it; 6Pediatric Neurology and Muscular Diseases Unit, IRCCS Istituto “Giannina Gaslini”, Via Gerolamo Gaslini 5, 16147 Genoa, Italy; riva.anto94@gmail.com (A.R.); dottorpiccolo.gianluca@gmail.com (G.P.); strianop@gmail.com (P.S.); 7Department of Neurosciences Rehabilitation, Ophthalmology, Genetics, Maternal and Child Health (DiNOGMI), University of Genoa, Largo Paolo Daneo 3, 16132 Genoa, Italy; scarsoacco@hotmail.com (A.A.); federico.zara@unige.it (F.Z.); 8Unit of Medical Genetics, IRCCS Istituto Giannina Gaslini, Via Gerolamo Gaslini 5, 16147 Genoa, Italy; 9Child Neurology and Neuropsychiatry Unit, Department of Human Pathology in Adult and Developmental Age “Gaetano Barresi”, University of Messina, Via Consolare Valeria 1, 98125 Messina, Italy; gdirosa@unime.it; 10Neonatal and Pediatric Intensive Care Unit, Department of Human Pathology in Adult and Developmental Age “Gaetano Barresi”, University of Messina, Via Consolare Valeria 1, 98125 Messina, Italy; egitto@unime.it; 11Department of Neuromuscular Diseases, UCL Queen Square Institute of Neurology and The National Hospital for Neurology and Neurosurgery, Gower Street, London WC1E 6BT, UK; e.cali@ucl.ac.uk (E.C.); s.efthymiou@ucl.ac.uk (S.E.); h.houlden@ucl.ac.uk (H.H.); 12Unit of Pediatric Nephrology and Dialysis, Department of Human Pathology in Adult and Developmental Age “Gaetano Barresi, University of Messina, Via Consolare Valeria 1, 98125 Messina, Italy; rchimenz@unime.it

**Keywords:** genotype, motor impairment, Nance-Horan syndrome, next-generation sequencing, congenital cataracts, dental anomalies, pediatric age

## Abstract

Nance-Horan syndrome (NHS) is a rare X-linked developmental disorder caused mainly by loss of function variants in the *NHS* gene. NHS is characterized by congenital cataracts, dental anomalies, and distinctive facial features, and a proportion of the affected individuals also present intellectual disability and congenital cardiopathies. Despite identification of at least 40 distinct hemizygous variants leading to NHS, genotype-phenotype correlations remain largely elusive. In this study, we describe a Sicilian family affected with congenital cataracts and dental anomalies and diagnosed with NHS by whole-exome sequencing (WES). The affected boy from this family presented a late regression of cognitive, motor, language, and adaptive skills, as well as broad behavioral anomalies. Furthermore, brain imaging showed corpus callosum anomalies and periventricular leukoencephalopathy. We expand the phenotypic and mutational NHS spectrum and review potential disease mechanisms underlying the central neurological anomalies and the potential neurodevelopmental features associated with NHS.

## 1. Introduction

Genetic brain developmental disorders with associated psychomotor regression include a broad variety of monogenic conditions with expanding clinical differential diagnosis, genetic heterogeneity, and associated disease mechanisms [1,2,3]. Despite being in the era of next-generation sequencing (NGS), the etiology and disease mechanisms underlying regressive neurodevelopmental impairment remain undetermined in a certain proportion of cases [4,5]. Defining the full spectrum of disease-causing molecular pathways underlying neurodevelopmental disorders will help to diagnose and monitor developmental trajectories in children affected with these conditions [6,7,8,9,10,11]

The neurodevelopmental condition known as ‘Nance-Horan syndrome’ (NHS) (OMIM 302350) is characterized by frequent intellectual disability and autistic features against a background of broad congenital anomalies, including congenital cataracts and dental abnormalities; distinctive facial features such as long and narrow face, anteverted pinnae, broad nose, and brachymetacarpia may also frequently occur in these patients [12,13,14,15,16,17,18]. Heterozygous females may present mild and variable clinical signs [19]. Various mutations in *NHS* and minor variations of the phenotypical features have been described [18,19,20]. NHS is caused by mutations in the *NHS* gene located on Xp22.13 [21], which is expressed in the midbrain, retina, lens, and tooth [22,23]. To date, the most frequently reported pathogenic mutations are either nonsense or frameshift mutations, which result in either nonsense mediated decay (NMD) of the respective mRNA or truncation of the respective protein [18,21]. In addition, a few microdeletions at Xp22.13, involving the *NHS* gene, have been also identified [17,22], some of which encompass also other genes, such as the *CDKL5* gene [24,25]. Here, we report a hemizygous stop mutation in the *NHS* gene (NM_001291867.2:c.375C>A; p.(Cys125Ter) detected by whole exome sequencing (WES) in a Sicilian boy affected with congenital bilateral cataracts and dental anomalies who displayed regressive brain developmental disorders with impairment of cognition and motor abilities in late childhood; also, brain imaging showed corpus callosum anomalies and periventricular leukoencephalopathy. This study further highlights the importance of WES in the diagnostic work-up of monogenic bilateral congenital cataracts.

## 2. Case Report

### 2.1. Patient Presentation

The boy was born at 27 weeks after a physiological pregnancy via cesarean delivery. He was the first child of healthy and unrelated parents (Figure 1A). The patient’s family history was unremarkable for intellectual disability and congenital anomalies. Apgar scores at 1 and 5 min were 4 and 7, respectively. Data regarding the perinatal history are limited because the child was born at a different care institution. At birth, the newborn was admitted to the neonatal intensive care unit for prematurity and required some ventilatory support in the first days of life. During the first week of life, systemic physical examination revealed bilateral lens opacities. Ophthalmologic examination confirmed the presence of bilateral congenital nucleo-cortical cataract, which underwent surgical correction in the first weeks of life. Other ocular features included microphthalmia, microcornea, astigmatism, nystagmus, and strabismus. Additional investigations, including ultrasound of the abdomen, echocardiogram, and infectious and metabolic (including galactosemia) work-up were all normal. Since his early infancy, the patient presented with diplegia. Cognitive and language development was reported as normal, as well as his social communication and interaction abilities. At the age of 2 years, brain magnetic resonance imaging (MRI) was performed and reveled an enlargement of the supratentorial portion of the ventricular system due to the enlargement of posterior horns, and middle cranial fossa arachnoid cysts were seen. These radiological findings were attributed at that time to possible perinatal ischemic injury due to prematurity. A standard motor rehabilitation program was set up. At 5 years of age, the child presented an episode characterized by generalized hypertonia, nystagmus, and bruxism, lasting a few minutes. Initially, low-grade fever and earache were considered as causative of the abovementioned symptoms. During the subsequent months, interactional skills started regressing and the development of obsessive-compulsive behavioral pattern was observed. At 8 years of age, a progressive loss of the previously acquired motor, language, and adaptive skills occurred, and the child exhibited psychic and/or motor agitation and either self-reported or not self-reported aggressive behavior. At the physical examination, the patient presented small teeth with abnormal implant (Figure 1B). Signs of vegetative nervous system activation, such as palmar and plantar hyperhidrosis and hypersalivation, were noticed. Electroencephalogram during wakefulness did not show significant abnormalities. A follow-up brain MRI scan was consistent with periventricular leucomalacia, with no more detectable enlargement of ventricular system. Risperidone, gamma-amynobutiric acid, and melatonin were started, with poor benefits. Autoimmunity work-up showed raised serum levels of anti-basal ganglia antibody (ABGA), but their dosage in the cerebrospinal fluid was negative. Autoimmunity investigation of cerebrospinal fluid showed non-specific results, finding the presence of antibodies directed towards antigens of 80 and 110 kDa, not yet identified and of uncertain pathogenetic role. Further autoimmunity (anti-nuclear, extractable nuclear antigens, lupus anticoagulant, anticardiolipin, and antiphospholipid antibodies—on serum—and anti-glutamic acid, anti-N-methyl-D-aspartate receptor, anti-Aquaporin 4, and anti-voltage-gated potassium channels—both on serum and liquor-) and metabolic (serum and urinary aminoacids, Fehling test, and ceruloplasmin) work-up were all normal. Mitochondrial encephalopathy with lactic acidosis and stroke-like episodes (MELAS syndrome) was ruled out based normal serum lactate and pyruvate levels, negative mitochondrial DNA testing, and absence of *POLG* intragenic variants in exome sequencing data. Karyotype and array-comparative genomic hybridization (CGH) were normal. A preliminary diagnosis of autoimmune encephalitis was hypothesized, and oral steroids for three months and intravenous immunoglobulin therapy were attempted, with no clinical improvement. At 9 years of age, a new MRI scan confirmed previous findings and additionally detected the deformation of both eyeballs at the rear portions (Figure 2).

### 2.2. Whole Exome Sequencing Results

After informed consent, whole exome sequencing (WES) was performed in the DNA extracted from the peripheral blood of the affected boy and his healthy parents. Nextera Rapid Capture Enrichment kit (Illumina) was used according to the manufacturer instructions. Libraries were sequenced on an Illumina HiSeq3000 using a 100-bp paired-end reads protocol. Sequence alignment with the human reference genome (UCSC hg19), variants calling, and annotation were performed as described elsewhere [26,27]. This led to the identification of a nonsense hemizygous variant in *NHS* (NM_001291867.2:c.375C>A; p.(Cys125Ter)), which is the causative gene for Nance Horan syndrome (NHS; Figure 1D). The variant in *NHS* emerged as the explanation of the disease pathogenesis given the consistency of the boy’s phenotype with NHS and the impact of the truncating variant identified by WES. This mutation is reported as pathogenic in ClinVar database (ClinVar ID: VCV000850838.2; http://www.clinvar.com (accessed on 19 July 2021)). No additional cases either carrying this particular *NHS* variant or presenting overlapping developmental regressive features were identified by a screening of available matchmaker platforms such as Genematcher (https://genematcher.org (accessed on 19 July 2021)), Variant Matcher (https://variantmatcher.org (accessed on 19 July 2021)), and DDD (https://www.deciphergenomics.org/ddd (accessed on 19 July 2021)). No additional pathogenic mutations or biallelic (compound heterozygous, homozygous) or de novo variants in known disease-causing genes emerged from the analysis of the WES data from the family.

### 2.3. Expression and Protein-Protein Interaction Analyses

To better understand the central neurological anomalies and the neurodevelopmental (regressive) features observed in our NHS family, we also explored the mRNA brain expression data as well as potential interactors of NHS using publicly availably web-based tools and datasets. To this end, we first interrogated microarray data (Affymetrix Exon 1.0 ST) from human post-mortem brain tissue collected by the UK Human Brain Expression Consortium as previously described [28]. This analysis showed higher expression in the frontal and temporal regions (Figure 3A); the mouse brain expression from the Allen Brain Atlas revealed high NHS expression in the frontal regions and the corpus callosum, consistent with the imaging phenotype of our family (Figure 3B). Then, we explored potential interactors of NHS using the STRING database (https://string-db.org (accessed on 19 July 2021)), and this revealed a close association with CDKL5 (Figure 4). This protein is encoded by the *CDKL5* gene (MIM #300203), known to be implicated in neurodevelopmental disorders with possible regressive features.

Figure 3A Boxplots of NHS mRNA expression levels in ten adult brain regions (source: BRAINEAC, see: http://braineac.org (accessed on 19 July 2021)). The expression levels are based on exon array experiments and are plotted on a log2 scale (y axis). This dataset was generated with Affymetrix Exon 1.0 ST arrays and brain tissue originating from 134 control individuals, collected by the Medical Research Council Sudden Death Brain and Tissue Bank. This plot shows variation in NHS expression across the ten brain regions analyzed, such that expression is higher in the frontal and temporal cortex thank in other brain regions. Abbreviations are as follows: PUTM, putamen; FCTX, frontal cortex; TCTX, temporal cortex; OCTX, occipital cortex; HIPP, hippocampus; SNIG, sub-stantia nigra; MEDU, medulla (specifically the inferior olivary nucleus); WHMT, intralobular white matter; THAL, thalamus; and CRBL, cerebellar cortex. Figure 3B NHS mRNA expression in the mouse brain in sagittal section. NHS was highly expressed in the frontal cortex and the cerebellum. Images were obtained from the Allen Mouse Brain Atlas (Allen Institute for Brain Science, see: http://mouse.brain-map.org/gene/show/83372 (accessed on 19 July 2021)). Expression intensity is color-coded in Nissl staining.

Based on the STRING data (see: https://string-db.org (accessed on 19 July 2021)), NHS protein is predicted to interact with a number of other proteins, some of which are encoded by genes already implicated in neurodevelopmental disorders sometimes associated with regressive features (*CDKL5*, *TRAF7*).

## 3. Discussion

Genetic brain developmental disorders of paediatric age include a variety of conditions that are frequently monogenic and associated with expanding clinical differential diagnosis [28,29,30]. The underlying genetic causes and the disease mechanisms associated with these disorders are highly heterogeneous and yet not completely understood. However, in recent years, NGS technologies (including exome and genome studies) revealed an increased complexity underlying a large proportion of infantile brain developmental disorders with variably associated congenital anomalies [30,31,32]. Many of the gene discovery in this field led to immediate benefits in terms of refining clinical phenotypes, valuable prognostic information, and potential targeted therapies in children with neurodevelopmental disorders [33,34,35,36].

The NHS-related neurodevelopmental disorder was first described by Margaret Horan and Walter Nance in 1974 [37,38]. In 1990, the causative gene was located within the region Xp21.1-Xp22.3 [39], and in 2003 the *NHS* gene was identified [21]. NHS is inherited in a X-linked semidominant pattern with high penetrance. The *NHS* gene consists of 10 coding exons and encodes three different isoforms (*NHS* A–C) as a result of alternative splicing [21,40]. Isoform A (*NHS*-A) is the most important transcript in NHS pathogenesis. It is expressed in multiple tissues, including the lens, brain, craniofacial mesenchyme, and dental primordial, where it regulates morphology and development [22,23]. To date, at least 44 mutations in *NHS* have been reported (Human Gene Mutation Database. Available from http://www.hgmd.cf.ac.uk/ac/gene.php?gene=NHS (accessed on 19 July 2021)), in absence of conclusive phenotype–genotype correlations [13]. NHS presents with varied expression among affected males and heterozygous females. The majority of the published studies focus on congenital cataracts, which are detected shortly after birth and result in profound visual loss if left untreated. Here, the case of a boy who presented bilateral congenital cataracts and a regressive neurodevelopmental disorder that led to late-onset cognitive and motor disability is described.

Some neuropsychological alterations are described in NHS subjects, and about 30% of patients with NHS presented intellectual disability; also, delayed psychomotor development, sleep–wake rhythm disorders, autism, aggression, anxiety, and stereotypical behavior are described [21].

Not all the phenotypic traits previously reported in males with NHS were present in our Proband, such as facial dysmorphisms, brachymetacarpia, and congenital cardiac defects. Additionally, the patient showed a complex central neurological involvement, characterized by a late regression of cognitive and language abilities and progressive motor impairment. This was not reported before in the spectrum associated with this condition and could be related to the implication of *NHS* in late brain adaptive development. Phenotypic heterogeneity associated with this disease-causing p.Cys125Ter variant or with the previously reported wide neurological features often described in the context of the NHS clinical spectrum may also represent the effects of modifying genes and stochastic processes during development.

Although the central anomalies identified by brain MRI and characterized by corpus callosum dysgenesis (and periventricular leukoencephalopathy) remain largely unexplained, the high expression of *NHS* in certain brain regions (e.g., fronto-parietal cortex, corpus callosum) supports a potential role of this gene in brain development and function.

Considering the clinical history of our patient, it is not possible to establish a certain correlation between the mutation in the *NHS* gene found in the proband and his phenotypic characteristics. It is possible that additional genetic and non-genetic factors may have influenced the neurological alterations exhibited by the patient, particularly the history of extreme prematurity and a possible autoimmune disease, although immunomodulant treatment was not effective in controlling the clinical symptoms.

Revision of additional brain imaging studies from individuals affected with NHS may clarify whether these anomalies should be considered as part of the neurological spectrum of this condition or as coincidental associations. Congenital cataracts were the very first sign of the syndrome in our case and represent the clinical hallmark of NHS in most affected individuals. Congenital cataracts may have environmental causes or contributions (e.g., TORCH infection, perinatal trauma, drug, or chemical exposure), but a monogenic basis should always be carefully considered in these children [41]. Wide range of investigations are often performed to identify a molecular cause underlying congenital cataracts; these include TORCH screening; karyotyping; urinalysis for reducing agents and organic amino acids; and measurement of plasma galactokinase levels and basic blood tests, such as full-blood count and liver function tests [42]. Nonetheless, current standard investigation for congenital cataracts is of low clinical utility in most cases as a means of identifying a diagnosis, and the time taken to diagnosis has been shown to be disappointing. Accordingly, our Patient underwent a range of investigative tests with no diagnostic return. Genetically determined cataracts may be isolated or syndromic and have been linked to mutations in over 110 genes [41]. In childhood, there are many syndromes with congenital cataracts as a phenotypic feature [43], which may challenge the healthcare providers during the early diagnostic process. A recent study by Musleh et al. [41] showed that NGS can deliver positive diagnosis in most of patients with congenital cataracts, if implemented in the diagnostic pathway by an integrated pediatric genetic team. NGS represents an effective methodology for the diagnosis of single-gene disorders causing bilateral congenital cataracts, identifying the cause in 60% up to 70–80%, respectively, in clinical [41] and research [44] settings. NGS cataract panel testing allowed one to state a diagnosis for our patient’s pathologic features. NGS testing improved diagnostic rates and time to diagnosis, as well as changing clinical management. NGS has not to be considered as a substitute for array-CGH in cases where there are cataracts plus additional malformations, growth, or developmental problems [43,44].

NHS was also known as “cataract-dental syndrome”. Burdon et al. [21] have demonstrated that *NHS* gene is expressed in the cap stage of the primordial tooth, that is, a very early stage of odontogenesis, involving alteration of crown morphology and the number of teeth that have evolved. In accordance, the dental anomalies in the NHS are present both in the primary and permanent dentition but might be less evident in the primary one. Dental abnormalities usually consist of screwdriver-shaped incisors, which is consistent with the original description [37], supernumerary maxillary incisors, and diastema [45]. In our patient, teeth morphology has been focused on later in the diagnostic work-up, after the mutation has been detected by NGS. The combination of congenital cataracts and teeth anomalies (especially screwdriver-shaped incisors and bud-shaped molars) represents a strong clinical indication of NHS. An increased knowledge and awareness of these peculiar morphological alterations might be an early diagnostic clue to NHS. The inclusion of a dental examination by dental professionals might be of value in an early stage of the diagnostic process [45].

In summary, we reported a boy affected with NHS, expressed with congenital bilateral cataracts, teeth anomalies, and a severe late regression of his neurological development, affecting multiple domains (e.g., cognition, motor development, language, and behavior). Additionally, he presented some central midline anomalies such as corpus callosum dysgenesia as well as white matter involvement on his brain imaging. We suggest that patients with congenital cataracts-related disorder should be candidate for NGS and clinical WES testing to avoid the occurrence of late diagnoses and improve follow-up of these families. Additionally, a careful examination of neurological and neuroradiological features is important in the context of NHS, and further studies will be needed to precisely assess the impact of the *NHS* gene disruption in late brain developmental processes and adaptive neurological skills. The identification of abnormal dentition and congenital cataracts in early infancy should raise clinical suspicion of NHS, and a timeline genetic diagnostic work-up and a close neurological follow-up through ages may be of pivotal importance in these families. The identification of further affected individuals carrying *NHS* mutations and presenting incomplete aspects of the clinical syndrome as well as unusual neurological and neurological features will help to better refine genotype–phenotype correlations associated with the NHS spectrum and, thus, to improve clinical management and follow-up strategies.

## Figures and Tables

**Figure 1 brainsci-11-01150-f001:**
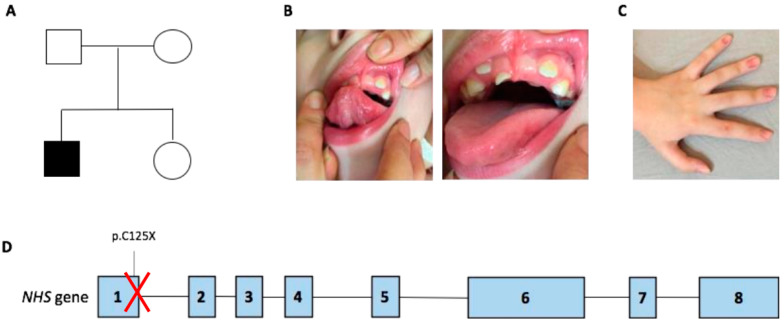
(**A**) Pedigree of the family. (**B**) Dental anomalies of the patient, including notching of incisors and supernumerary teeth (right and left panel). (**C**) Note the mild dyschromia of sub-ungual space. (**D**) Schematic representation of the *NHS* gene with the p.C125X hemizygous variant identified in the exon 1 of the gene.

**Figure 2 brainsci-11-01150-f002:**
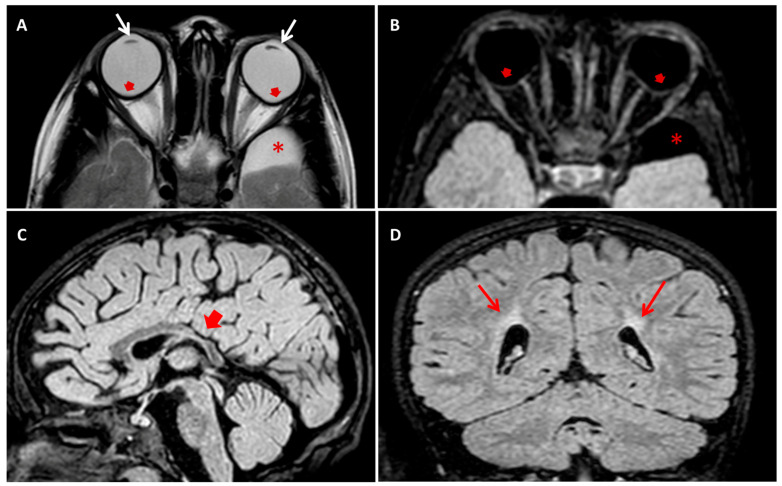
(**A**) Axial T2 turbo-spin echo (TSE) and (**B**) axial T2 fluid attenuated inversion recovery (FLAIR) brain MRI images showing bilateral crystalline lens thinning (white arrows in (**A**)) and circumscribed outpouching along the posterior aspect of the ocular bulbs, temporal to the optic disc (red arrows in (**A**,**B**)). Note a well circumscribed left temporo-polar arachnoid cyst (* in (**A**,**B**)). (**C**) Sagittal T2 FLAIR and (**D**) Coronal T2 FLAIR brain MRI images showing thinning of posterior trunk of corpus callosum (red short arrow in (**C**)) and bilateral deep paratrigonal white matter hyperintensity due to periventricular leukoencephalopathy (red arrows in (**D**)).

**Figure 3 brainsci-11-01150-f003:**
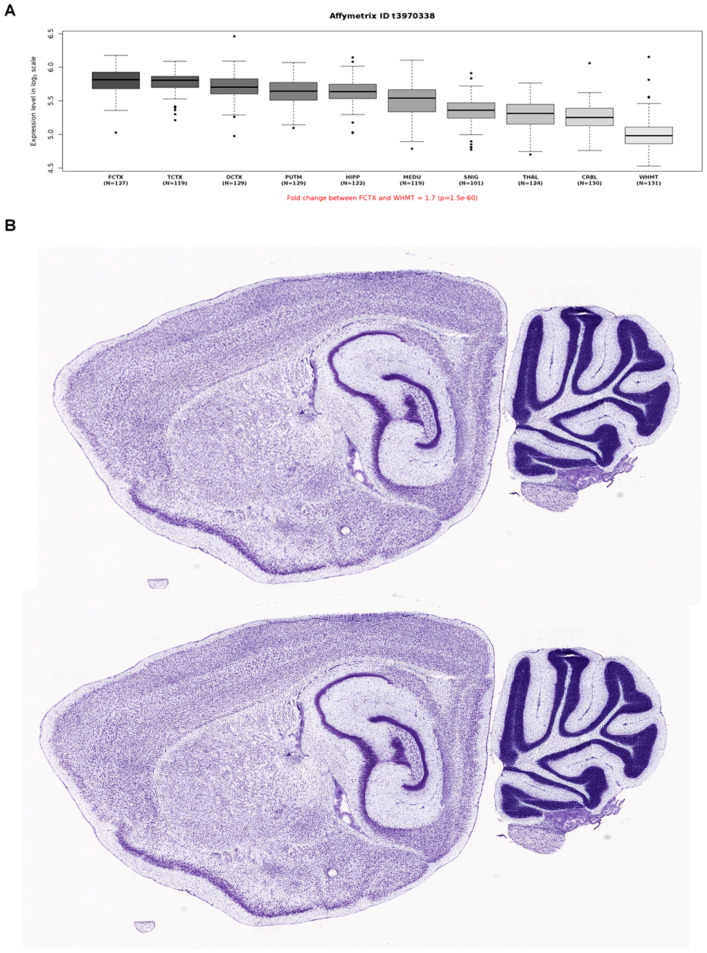
Summary of *NHS* gene expression in human (**A**) and mouse (**B**) brain.

**Figure 4 brainsci-11-01150-f004:**
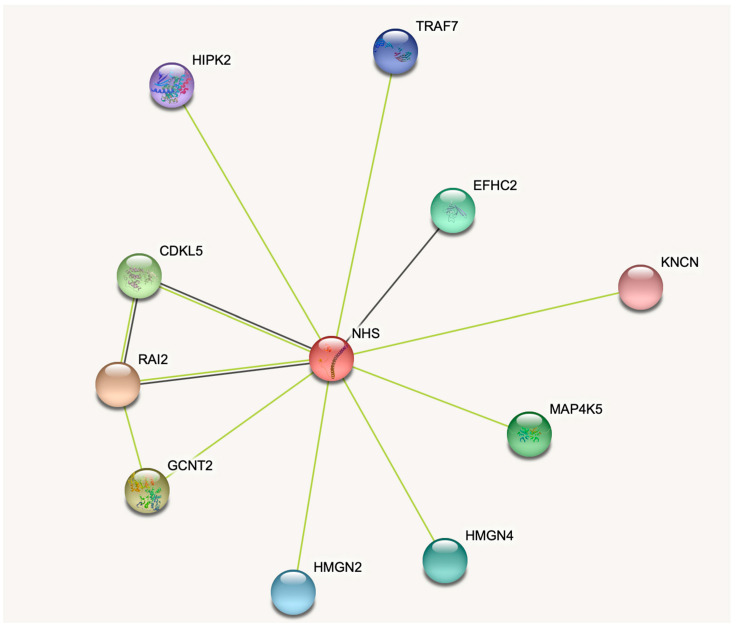
General protein–protein network showing different interactions of candidate genes with the NHS protein.

## Data Availability

Not applicable.

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
