# Peer review of "Prominent and Regressive Brain Developmental Disorders Associated with Nance-Horan Syndrome"

_brainsci, 2021, doi:10.3390/brainsci11091150_

Round 1
Reviewer 1 Report
Casto, et al. present a fascinating case of Nance-Horan Syndrome demonstrating developmental regression and previously undescribed neurological features. They draw interesting parallels to patients with mutations in CDKL5, a protein in the same pathway and prompt reconsideration of what is known about NHS, a disorder that was first described in the 1970s. However, some concerns limit my overall high enthusiasm for the paper.
- Developmental regression would be an important and novel finding for NHS. However, the history of extreme prematurity and possible neurologic autoimmunity make it difficult to be certain whether this can truly be attributed to NHS. The authors should describe any efforts to reach out to the NHS community or physician/researcher networks (such as GeneMatcher) to determine whether other NHS patients have this phenotype. The authors should also mention these caveats as limitations of their study in the Discussion
- The perinatal history needs to be described in more detail, as it may have contributed to at least some of the MRI findings described in this patient
- The text is not clear; was ABGA assay in CSF?
- How was MELAS "ruled out?"
- The latest HGVS nomenclature recommendations should be utilized to describe genomic variants
- The Discussion should include more about the neurological features in the context of what is already known
- The authors mention phenotypic heterogeneity but do not discuss the current mutation in this context
Reviewer 2 Report
The authors describe a 9-year-old boy with late regression of cognitive, motor, language, and adaptive skills, as well as broad behavioral anomalies in addition to congenital, bilateral cataract. Brain imaging revealed corpus callosum anomalies and periventricular leukoencephalopathy. Whole exome sequencing in the index patient identified a novel premature termination codon in exon 1 of the NHS gene. The authors conclude that their findings expand the phenotypic spectrum of NHS mutations. No additional pathogenic mutations or de novo variants in known disease-causing genes were found in the WES data from the family. Karyotype and array-comparative genomic hybridization (aCGH) were normal.
Initially, cognitive and language development were reported as normal, as well as social communication and interaction abilities of the index patient. Magnetic resonance imaging (MRI) reveled an enlargement of the supratentorial portion of the ventricular system. Also, middle cranial fossa arachnoid cysts were seen. Radiological findings were attributed to possible perinatal ischemic injury prematurity due to a perinatal ischemic injury. Autoimmunity examination of cerebrospinal fluid detected the presence of antibodies directed towards unknown antigens of 80 and 110 kDa. A preliminary diagnosis of autoimmune encephalitis was stated, but oral application of steroid for three months and intravenous immunoglobulin therapy did not result in clinical improvement of the patient.
Comments to the authors
Major:
It is not clear whether or not all clinical findings can be attributed to the mutation in the NHS gene. Alternatively, a second trait (either genetic or environmental) might be present in the index patient. Possible other reasons may include perinatal ischaemia and/or autoimmune disease of the brain. Therefor the presented data do not support the conclusion that all clinical findings are really a straight consequence of the sequence variant in the NHS gene.
The variant described in the manuscript (NM_001291867.2(NHS):c.375C>A; p.(Cys125Ter)) is not novel. I found a ClinVar entry for this nucleotide substitution (ClinVar ID: VCV000850838.2).
Minor:
|
Line 6 |
Xp22.13.4? |
|
Line 71 |
Truncating mutations more frequently lead to nonsense mediated decay (NMD) of the respective mRNA instead of truncated proteins, unless there is compelling evidence for the latter. |
|
Line 73 |
Since the NHS gene is on the X chromosome, the variant does not occur homozygous in the affected boy but rather hemizygous. |
|
Line 74 |
Sequence variant nomenclature is wrong! (NHS:NM_001291867:exon1:c.C375A;p.C125X, NHS:NM_198270:exon1:c.C375A;p.C125X) Use the official nomenclature according to the Human Genome Variation Society (https://varnomen.hgvs.org). |
|
Line 85 |
Do not use mental retardation. |
|
Line 109 |
Alterations instead of alternations? |
|
Lines 134 & 135 |
Variant nomenclature (see above). |
|
Line 144 |
rmRNA? |
Round 2
Reviewer 1 Report
The authors have satisfactorily addressed no concerns
Author Response
Thank you for your comments and positive considerations.